# The PA Subunit of the Influenza Virus Polymerase Complex Affects Replication and Airborne Transmission of the H9N2 Subtype Avian Influenza Virus

**DOI:** 10.3390/v11010040

**Published:** 2019-01-09

**Authors:** Mengchan Hao, Shaojie Han, Dan Meng, Rong Li, Jing Lin, Meng Wang, Tong Zhou, Tongjie Chai

**Affiliations:** 1College of Veterinary Medicine, Shandong Agricultural University, 61 Daizong Street, Taian 271018, China; mengchan1993@126.com (M.H.); 18763896230@163.com (S.H.); iamli_z@126.com (D.M.); lirong19900129@163.com (R.L.); 18763806701@163.com (J.L.); 18854937499@163.com (M.W.); 13864453175@163.com (T.Z.); 2Collaborative Innovation Center for the Origin and Control of Emerging Infectious Diseases, Taishan Medical University, Taian 270016, China

**Keywords:** H9N2 AIV, pandemic 2009 H1N1 virus, reassortment, replication, airborne transmission

## Abstract

The polymerase acidic (PA) protein is the third subunit of the influenza A virus polymerase. In recent years, studies have shown that PA plays an important role in overcoming the host species barrier and host adaptation of the avian influenza virus (AIV). The objective of this study was to elucidate the role of the PA subunit on the replication and airborne transmission of the H9N2 subtype AIV. By reverse genetics, a reassortant rSD01-PA was derived from the H9N2 subtype AIV A/Chicken/Shandong/01/2008 (SD01) by introducing the PA gene from the pandemic influenza A H1N1 virus A/swine/Shandong/07/2011 (SD07). Specific pathogen-free (SPF) chickens and guinea pigs were selected as the animal models for replication and aerosol transmission studies. Results show that rSD01-PA lost the ability of airborne transmission among SPF chickens because of the single substitution of the PA gene. However, rSD01-PA could infect guinea pigs through direct contact, while the parental strain SD01 could not, even though the infection of rSD01-PA could not be achieved through aerosol. In summary, our results indicate that the protein encoded by the PA gene plays a key role in replication and airborne transmission of the H9N2 subtype AIV.

## 1. Introduction

The influenza virus belongs to the orthomyxoviridae RNA virus family, which contains six to eight segmented of linear, negative-sense, single-stranded RNA. There are four types of influenza viruses: influenza A, influenza B, influenza C, and influenza D [1,2,3,4]. Influenza A mainly infects poultry, and some strains are known to infect mammals and humans. Avian influenza is a form of respiratory infectious disease caused by a subtype of avian influenza virus of influenza A. The H9N2 subtype avian influenza virus (AIV) is a low-pathogenic avian influenza, which normally causes a slight drop in egg production, mild respiratory symptoms, and immune dysfunction with no significant pathological changes [5,6,7]. However, mixed infection of H9N2 with other pathogenic microorganisms increases the incidence of diseases, which leads to severe respiratory syndromes, a sharp drop in egg production, and a high mortality rate of 10% to 60% [6,8,9,10,11,12]. Hence, H9N2 poses a significant threat to public health and can cause serious economic loss to the poultry industry.

Reassortment is an important mechanism for producing new influenza viruses from multiple co-infected subtype influenza viruses by exchanging gene segments [13,14]. Among the four flu pandemics in the past, three were caused by reassortants during which the genes were derived from both human and avian influenza viruses [15,16]. Currently, H9N2 is the most prevalent subtype AIV in many countries and regions, including China. It can serve as a viral gene segment donor for the reassortant and is considered one of the candidates for generating pandemic influenza virus strains [17,18,19]. Meanwhile, the new pandemic influenza A H1N1/2009 virus has been circulating worldwide in humans and mammals [20,21,22]. Therefore, co-infection of these two subtypes in the same host (such as pigs) may lead to a high occurrence of reassortment. The PA gene of the new pandemic influenza A H1N1/2009 virus may play an important role in aerosol-mediated transmission across populations [23].

The most common mammalian model used for influenza research, ferrets, are large and expensive. As a result, the facilities required to house these animals are not widely available and ferrets are not available from most laboratory animal suppliers. Conversely, guinea pigs are relatively small and easy to handle. Meanwhile, they are susceptible to the influenza virus. For these reasons, guinea pigs have been used in evaluating the replication and transmission of the influenza virus. In this study, we constructed a reassortant rSD01-PA by replacing the PA gene of the H9N2 subtype AIV SD01 with that from the new pandemic influenza A H1N1/2009 virus. We selected SPF chickens and guinea pigs as the animal models for reassortant replication and airborne transmission studies.

## 2. Materials and Methods

### 2.1. Virus Strains, Cells, and Animals

A/Chicken/Shandong/01/2008 H9N2 strain (SD01), A/Swine/Shandong/07/2011 H1N1 strain (SD07), human embryonic kidney (293T) cells, and Madin-Darby Canine Kidney (MDCK) cells were provided by Shandong Agricultural University Environmental Microbiology Laboratory. The viruses were passaged in nine-day-old specific pathogen-free (SPF) chicken embryos. The 293T and MDCK cells were cultured in Dulbecco’s minimum essential medium (Gibco-BRL, Grand Island, NY, USA) supplemented with 10% fetal bovine serum, penicillin (100 unit/mL), and streptomycin (100 mg/mL), and incubated at 37 °C with 5% CO_2_. The SPF chickens were purchased from the Poultry Farm of Shandong Academy of Agricultural Sciences (Jinan, Shandong, China) and the SPF guinea pigs were purchased from the Shandong Taibang Biological Products Company Limited (Tai’an, Shandong, China). All animal experiments were handled in accordance with the guidelines of the Shandong Agricultural University Animal Care and Use Committee (No. SDAUA-2016-004).

### 2.2. Construction of Plasmids and Rescue of the Reassortant

The target segment was amplified using a reverse transcription-polymerase chain reaction (RT-PCR) with primers designed according to homologous reassortment. The linearized vector pHW2000 digested and ligated the purified segment. The resulting plasmid was transformed into DH-5α competent cells. Seven positive plasmids for the SD01 strain and one PA positive plasmid of the SD07 strain were obtained and sequenced. SD01 and SD07 plasmids (0.5 μg each) dissolved in OPTI-MEMI were mixed with Lipofectin reagent (Invitrogen, Carlsbad, CA, USA) for 30 min. The mixture was used for co-transfection in 293T cells.

The 293T cells were incubated in a 5% CO_2_ incubator at 37 °C for 6 h, before replacing the transfection solution with 2 mL fresh OPTI-MEMI containing 2 μg/mL TPCK-trypsin. The cells were incubated for another 48 to 72 h before inoculating 9- to 11-day-old SPF chicken embryos for viral replication. The rescued recombinant virus was named rSD01-PA. The rescued viral RNA was extracted. The DNA was synthesized via reverse transcription and amplified for sequencing to verify the introduced PA gene. Then, the Reed-Muench method was used to calculate the 50% egg infective dose (EID_50_).

### 2.3. Replication Ability of Viruses in the Lung Tissues of SPF Chickens

Studies using H9N2 low-pathogenic AIV were conducted in a Biosafety Level 2+ laboratory approved by the China National Accreditation Service for Conformity Assessment. All animal studies were carried out in strict accordance with the guidelines of Laboratory Animal Management by the National Council for Science and Technology.

The parental strain SD01 and the reassortant rSD01-PA were diluted to 10^6^–10^8^ EID_50_. Each diluted sample was used to inoculate five 4-week-old SPF chickens. The clinical symptoms, such as chicken’s mental state, were observed and recorded. The lung tissues of three SPF chickens were collected at 5 days post inoculation (dpi) and 0.5 g of lung tissues were ground in 1 mL sterilized phosphate buffer solution (PBS). A total of 0.2 mL of the supernatant diluted by 1000 times was used to inoculate the embryos of 9-day-old SPF chickens, which were placed in an incubator at 37 °C for 72 h. The allantoic fluid of each chicken embryo was collected under sterile condition, and the viral titer of each tissue was measured. Viral titers are expressed as mean logEID_50_/g of wet tissue ± SD. The other two SPF chickens were observed for 14 days, and then the serum was separated for determination of the antibody titer by the Hemagglutination/Hemagglutination inhibition (HA/HI) test with reference to the OIE standard.

### 2.4. Viral Transmission among SPF Chickens

Transmission experiments of influenza viruses among SPF chickens were carried out for the H9N2 subtype AIV SD01 and the reassortant rSD01-PA. Transmission experiments were carried out in 2100 mm × 800 mm × 1500 mm positive and negative pressure isolators, A and B, respectively (Figure 1A), which were connected via a sealed plastic tube (150 cm in length and 8 cm in diameter). The positive air compressor was adjusted to allow outside air to enter isolator A through a filter and then enter the isolator B through the sealed tube. The air flow was set at 0.05 to 0.2 m/s. The isolators were maintained at an appropriate temperature and humidity level. Thirty 4-week-old SPF chickens were randomly and evenly divided into three groups. The inoculation group was inoculated with 100 μL 10^6^ EID_50_ virus solution by eye and nasal inoculation. The other two groups were the direct contact group and the aerosol-mediated group, respectively. The inoculation group was placed in isolator A after inoculation. After 24 h, the direct contact group and the aerosol-infected group were placed in isolators A and B, respectively. The chickens were fed in strict accordance with the standards for the care of SPF chickens. The drinking water and feed were sterilized before use. Oropharyngeal and cloacal cotton swab samples were constantly collected at 2-day intervals and inoculated in SPF embryonated chicken eggs for observing virus shedding. Meanwhile, the serum samples and the air samples were collected.

Two AGI-30 liquid samplers [24] were used to collect air samples, which were then analyzed using the 7500 System SDS Software Version 1.2 by real-time RT-PCR. The sampler, which contained 20 mL PBS, was fixed 1 m above the ground at the sampling point and operated continuously for an optimized time of 30 min at an airflow rate of 12.5 L/min [25]. Samples were then stored at 4 °C and tested within 24 h. After centrifugation at 10,000× *g* for 45 min at 4 °C to remove bacteria and dust particles, the sample supernatant was ultra-centrifuged at 100,000× *g* for 2 h at 4 °C to collect the viral pellet, which was then re-suspended in 1 mL PBS.

### 2.5. Replication Ability of the Viruses in Guinea Pigs

To measure the pathogenicity of the two strains (SD01 and rSD01-PA) in guinea pigs, five 250 g female guinea pigs were selected, and 0.05 mL Sumianxin II was administered via intramuscular injection. About 10 min after injection, guinea pigs entered the anesthesia state. Then, 300 μL 10^6^ EID_50_ virus solution was administered via intranasal inoculation. The pigs were closely monitored until waking up. Clinical symptoms and mortality rates were recorded. On day 5 after inoculation, 0.5 g samples of the brain, turbinate, trachea, lung, and kidney tissues of three guinea pigs were collected from each group and ground after adding 1 mL sterilized PBS. The supernatant after centrifuge was diluted 10-fold and inoculated into the MDCK cells. The viral titer was determined for each tissue. Viral titers are expressed as mean logTCID_50_/g of wet tissue ± SD. The other two guinea pigs were observed for 14 days, and then, the serum was collected for determining antibody titers by the HA/HI method.

### 2.6. Transmission of Viruses among Guinea Pigs

Transmission experiments of influenza viruses among guinea pigs were carried out in 1450 mm × 800 mm × 1250 mm positive and negative pressure isolators, A and B (Figure 1B). The two isolators were connected via a sealed plastic tube (150 cm in length and 8 cm in diameter). The positive air compressor was adjusted to allow outside air to enter isolator A through a filter and then enter isolator B through the sealed tube. The air flow was set at 0.05 to 0.2 m/s, and the isolators were maintained at an appropriate temperature and humidity levels. Fifteen 250 g guinea pigs were randomly and evenly divided into three groups. The inoculation group was inoculated with 300 μL 10^6^ EID_50_ virus solution via nasal inoculation. The other two groups were the direct contact group and the aerosol-mediated group, respectively. The inoculation group was placed in isolator A after inoculation. After 24 h, the direct contact group and the aerosol-mediated group were placed in isolators A and B, respectively. The guinea pigs were fed in strict accordance with the standards for the care of guinea pigs. The drinking water and feed were sterilized before use. Nasal washing samples were constantly collected at 2-day intervals and titrated in MDCK cells. Meanwhile, the serum samples and air samples from the two isolators were constantly collected and tested as described in Section 2.4.

### 2.7. Statistical Analysis

The mean values were calculated using Microsoft Excel, the data were expressed as means ± standard deviation and were analyzed with Student’s t test. All statistical analyses were carried out using the SPSS v19.0 software package (version 19.0; SPSS Inc., USA). *p* < 0.05 was considered to be significant.

## 3. Results

### 3.1. HA Test and EID_50_ Determination

Reassortant rSD01-PA was successfully rescued by reverse genetics, which is indicated by the fact that the PA gene was from the new H1N1/2009 strain SD07, and the other seven genes were from SD01. The rSD01-PA was inoculated into nine-day-old SPF chicken embryos, which showed stable reproduction and proliferation after three generations. The HA test results were not significantly different between rSD01-PA and SD01 (*p* > 0.05, Table 1). The EID_50_ was also not significantly different between the two viruses (*p* > 0.05).

### 3.2. Replication Ability of Viruses in the Lung Tissues of SPF Chickens

Viruses were detected in all SPF chickens inoculated at various doses, and none of the chickens died of infection. The SPF chickens showed no clinical symptoms at the inoculation dose of 10^6^ EID_50_. No observable clinical disease was found in chickens except a slight inappetence and inactivity at the inoculation dose of 10^7^ EID_50_ virus. However, they recovered after one week. The viral titers in the lung tissues of the SPF chickens inoculated with the same dose did not differ between the two viruses (*p* > 0.05, Table 2). The viral titers did increase with the increase of the dose. The other two SPF chickens were tested for anti-H9 antibody at 14 dpi, and the results showed that all were positive, which indicates that the replacement of the viral PA gene did not affect its replication within the lung tissues of the SPF chickens.

### 3.3. Transmission of Viruses among SPF Chickens

#### 3.3.1. Oropharyngeal and Cloacal Cotton Swab Tests in SPF Chickens

Table 3 shows that the viruses reached the highest concentration at 6 to 10 dpi. At 2 dpi, viral replication could be detected in both the rSD01-PA and SD01 inoculated chickens, and the isolation rate was 60% and 20%, respectively. The SD01 viruses were detected in direct contact between 2 to 12 dpi. Similarly, viruses were also detected in the aerosol-mediated group between 6 to 12 dpi. The rSD01-PA virus was detected in the direct contact group between 6 to 12 dpi, but it was not detected in the aerosol-mediated group.

#### 3.3.2. Antibody Levels in SPF Chickens

Serum antibodies in the SPF chickens were not detected before inoculation. However, in the inoculation groups for both viruses (SD01 and rSD01-PA), the serum AIV antibodies were detected between 7 to 21 dpi, which showed an upward trend (Table 4) and indicated that both strains can infect SPF chickens and produce the corresponding AIV antibody. The SD01 AIV antibody was detected at 7 to 21 dpi and 14 to 21 dpi for the direct contact group and aerosol-mediated group, respectively, which indicates that SD01 can infect SPF chickens and induce antibody production through either direct contact or aerosol. However, the rSD01-PA AIV antibody was detected only in the direct contact group between 14 to 21 dpi. Antibody titers for the direct contact group had reduced compared with SD01 (*p* < 0.05, Table 4). It was not detected in the aerosol-mediated group, which indicates that the reassortant rSD01-PA after the PA gene replacement did not have the ability to form an aerosol-transmitted virus that could infect SPF chickens.

#### 3.3.3. Aerosol of the SD01 and rSD01-PA in the Isolators of SPF Chickens

The concentrations of the airborne H9 subtype AIV in the isolators were measured by fluorescence quantitative PCR after the air samples were collected and processed at 2, 4, 6, 8, 10, 12, and 14 dpi. The H9 subtype AIV could be detected in the air samples taken from the SD01 group at 4 dpi (Figure 2), and the concentration reached the peak value (5.31 × 10^4^ copies/m^3^ air) at 8 dpi, which is consistent with results showing that SD01 reached its peak at 8 dpi (Table 2). Yao et al. (2011) showed that SPF chickens can be infected when the air concentration of H9N2 AIV exceeds 4.88 to 7.2 × 10^3^ copies/m^3^ air [26]. However, the airborne H9 subtype AIV in the air samples was not detected in the group for rSD01-PA. Results reveal that rSD01-PA could not be transmitted via aerosol, which indicates that the virus without the PA gene of SD01 loses the ability to transmit via aerosol in SPF chickens.

### 3.4. Replication Ability of the Viruses in Guinea Pigs

The SD01 strain was not detected in guinea pigs after inoculation, which indicates that SD01 did not have a replication ability in guinea pigs. However, rSD01-PA was detected in the trachea and turbinate after inoculation with viral titers of 3.76 log10TCID_50_/g ± 0.64 and 4.32 log10TCID_50_/g ± 0.48, respectively. The virus titer in turbinate was higher (*p* < 0.05, Table 5), which indicated that the replacement of the PA gene enables rSD01-PA to infect guinea pigs and replicate in its turbinate and trachea. However, viral replication was not detected in the lungs, which could be related to the distribution of different sialic acid receptors. The other two guinea pigs were observed for 14 days during which the titers of the AIV antibodies in the serum were determined, and the AIV antibodies were detected in guinea pigs inoculated with rSD01-PA. Guinea pigs inoculated with the SD01 parental strain did not have the AIV antibody (Table 5).

### 3.5. Transmission of rSD01-PA among Guinea Pigs

#### 3.5.1. Detection of rSD01-PA in Guinea Pig Nasal Washes

Viruses could be isolated from the nasal washes of the inoculated and direct contact groups between 4 and 12 dpi, which shows that viral replication at 4 dpi lasted eight days (Table 6). At 4 dpi, rSD01-PA was detected in guinea pigs inoculated with rSD01-PA in both the inoculation and direct contact groups. The virus isolation rate was 60% in the inoculation group, and the virus isolation rate was 40% in the direct contact group at 4 dpi. Virulence continued until 12 dpi. However, the virus was not detected in the nasal washes from the aerosol-mediated group, which indicates that, after the PA gene replacement, rSD01-PA can infect and replicate in guinea pigs and can transmit via direct contact, but not via aerosol transmission.

#### 3.5.2. Detection of AIV Antibody Levels in Guinea Pigs

Before inoculation, the AIV antibody was not be detected in guinea pig serum. The AIV antibody was detected between 7 and 21 dpi and between 14 and 21 dpi in the inoculation group and the direct contact group, respectively (Table 7). However, the antibody was not detected in the aerosol-mediated group, which indicates that the reassortant rSD01-PA can infect guinea pigs to produce the corresponding antibody, and the infected guinea pigs transmit the virus via direct contact with the guinea pigs in the direct contact group and produce the AIV antibody. However, guinea pigs could not be infected via aerosols. The concentrations of the airborne H9 subtype AIV in the isolators were measured using fluorescence quantitative PCR after air samples were collected and processed at 2, 4, 6, 8, 10, 12, and 14 dpi. The virus was not be detected in air samples. Results show that, although rSD01-PA can infect guinea pigs, it did not have the ability to form an aerosol-transmitted virus.

## 4. Discussion

The influenza virus has many subtypes. During a mixed infection, reassortment and mutation are likely to occur during viral replication and proliferation, which leads to the production of new viruses, some of which may form new epidemic strains by converting from a low-pathogenic avian influenza to highly pathogenic influenza viruses [27,28,29]. Since 1918, many incidents of H1N1 infection have been reported, which led to severe illness and death. During the 2009 outbreak of a new influenza A H1N1, a rapid spread of the virus around the world brought high impacts on people’s psychology and behavior, and caused panic and anxiety, even though the lethality was relatively low compared with those in previous outbreaks of H1N1. Pigs can be infected by both the H9 subtype AIV and the new influenza [23,30,31], which can serve as a host for the reassortment of these two viruses. Therefore, new viruses with potential pandemic threats can be produced in pigs [32]. The scientific community has concerns that the new strains produced by reassortment or mutations can infect mammals and humans, which are potentially capable of causing pandemics. It is generally believed that it is only a matter of time before the virus adapts and spreads to humans.

Avian flu is a zoonotic infectious disease. The three aspects of the epidemic include the source of infection, the transmission route, and the susceptible animals. Influenza viruses are transmitted primarily through direct contact with contaminants or secretions, digestive tracts and aerosols, among which aerosols can travel long distances rapidly and efficiently, which makes it difficult to control and avoid [26]. If the highly pathogenic influenza virus can be effectively transmitted through aerosols in the host, it will pose a significant threat to public health [33].

To study the reassortment virus of the H9N2 subtype AIV and the new 2009 influenza A, we constructed seven transcription/expression plasmids of the H9N2 subtype AIV SD01 and one PA transcription/expression plasmid of the new influenza A virus SD07 strain by homologous reassortment. By reverse genetics, a recombinant virus rSD01-PA was obtained by replacing the PA gene of the H9N2 subtype AIV SD01 with that from the novel influenza A H1N1/2009 virus strain (SD07). In this study, SPF chickens and guinea pigs were selected as the animal models, which represent poultry and mammals, respectively. Infections of the virus among animals were carried out in positive and negative pressure isolators to study the replication abilities and airborne transmission of recombinant viruses.

Our studies show that replacement of the viral PA gene did not affect its replication capacity in chicken embryos and in the lung tissues of SPF chickens. The parental strain SD01 could not infect guinea pigs, but the rSD01-PA could effectively replicate in their turbinates and tracheas. The reassortant rSD01-PA lost airborne transmission ability among SPF chickens because of the single substitution of the PA gene. In guinea pigs, it could be spread through direct contact, but the parental strain SD01 could not. Infection with rSD01-PA could not be achieved through aerosols.

Avian influenza virus belongs to the family of orthomyxoviridae, possessing a segmented RNA genome with eight negative-sense segments [34]. The viral polymerase complex, which is composed of the PB2, PB1, and PA subunits, is involved in determining viral host range, replication, and pathogenicity and plays an important role in transmission [35,36,37]. The PA protein is the third subunit of the influenza A virus polymerase complex. Polymorphisms in PA influencing pathogenicity and host adaptation have been identified throughout the PA protein [38]. In recent years, data have suggested that PA plays an important role in overcoming the host species barrier and host adaptation of AIV. The PA protein functions as an endonuclease and a protease, and also participates in the binding of the viral RNA (vRNA)/complementary RNA (cRNA) promoter [39,40]. PA can be transferred into nuclei using nuclear localization signals and regulates cRNA/vRNA synthesis [41,42,43]. In this study, the reassortant rSD01-PA was derived from an avian influenza virus with high tropism to avian cells. The PA gene of rSD01-PA was from the swine influenza virus. Naffakh et al. (2008) showed that the source of the viral polymerase gene directly determines its growth and replication activity in various host-derived cells [38]. Viruses with swine influenza-derived polymerase genes usually exhibit higher polymerase activity in mammalian cells while the polymerase activity is low in avian cells. Therefore, compared with the parental strains, although the replication ability of rSD01-PA in SPF chickens did not significantly change, the aerosol transmission was completely lost. The reassortant rSD01-PA could infect guinea pigs, but the parental strain SD01 could not.

The host-expressed receptors are important factors that affect the transmission of influenza viruses through the respiratory tract. Influenza virus infection is mediated via binding of the viral HA protein to terminally attached α(2,3)-linked or α(2,6)linked sialic acids on cell surface glycoproteins [44]. Gao et al. detected the specificity of the respiratory receptors in guinea pigs using a lectin immunohistochemistry test. Their results indicate that both the upper respiratory tract and tracheal epithelial cells express α(2,3)-linked and α(2,6)linked sialic acids with the former being the predominant ones in the lungs [45]. The reassortant rSD01-PA was derived from an avian influenza virus whose parental strain preferentially binds to the α(2,3)-linked SA receptor. Our results show that, in guinea pigs, the parental strain could not replicate, but the reassortant rSD01-PA could. The replication pattern of rSD01-PA seen in guinea pigs could be related to the specific expression of its lung receptor. The viral properties governing replication of influenza viruses are complex. Whether they are related to the specific expression of the receptor needs further verification.

In summary, our study demonstrates that the replacement of a single PA gene affected the replication and transmission of the H9N2 subtype AIV in SPF chickens and guinea pigs, which provides a reference for the prevention and control of influenza virus pandemics. To investigate, in depth, the mechanism of replication and transmission of the H9N2 subtype AIV, further studies about reassortment in the PA gene are also needed.

## Figures and Tables

**Figure 1 viruses-11-00040-f001:**
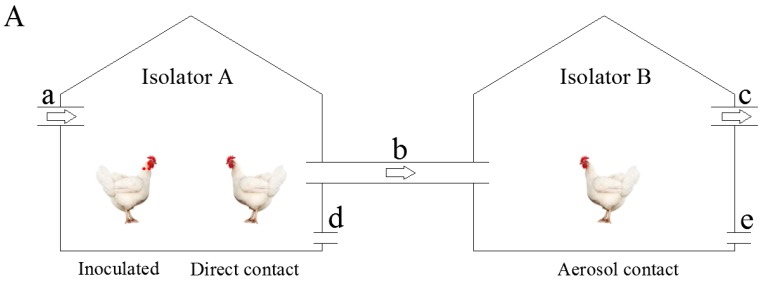
Transmission experiments of SPF chickens and guinea pigs. The experiments were constructed in a biosecurity level 2+ laboratory. (**A**). Ten inoculated SPF chickens and 10 direct contact SPF chickens were placed in the isolator A. Another 10 SPF chickens of aerosol contact were placed in isolator B, which was connected by an air tube with isolator A. (**B**). Five inoculated guinea pigs and five direct contact guinea pigs were placed in the isolator A. Another five guinea pigs of aerosol contact were placed in isolator B, which was connected by an air tube with A. (a) Positive pressure fan with High Efficiency Particle Air (HEPA) filters. (b) The tube connected the isolators and allowed the air flowing from isolator A to isolator B. (c) Negative pressure fan with HEPA filters (d) and (e) sampling holes from which air samples were collected.

**Figure 2 viruses-11-00040-f002:**
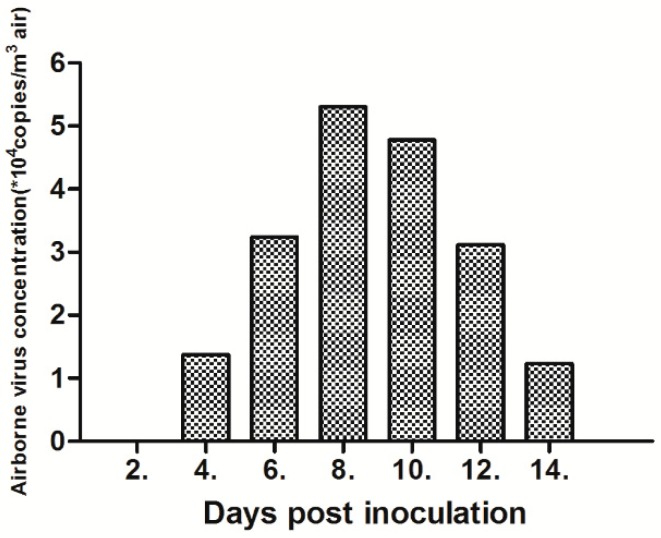
Virus concentration in the air of the isolators of SD01 for SPF chickens. The concentrations of the airborne H9 subtype AIV in the isolators were measured by fluorescence quantitative PCR after the air samples were collected and processed at 2, 4, 6, 8, 10, 12 and 14 dpi. The copies of the viruses were determined by the real-time RT-PCR method. Virus titers were not detected from the isolator of rSD01-PA, so they were not represented in this figure.

**Table 1 viruses-11-00040-t001:** Hemagglutination and infectious virus titers.

Virus	HA(log_2_ ± SD)	lgEID_50_/mL ± SD
SD01 ^a^	8.5 ± 0.13	9.31 ± 0.27
rSD01-PA ^b^	7.3 ± 0.34	8.27 ± 0.20

^a^ The parental strain A/Chicken/Shandong/01/2008 H9N2 strain (SD01). ^b^ The reassortant rSD01-PA was derived from the H9N2 subtype AIV SD01 by introducing the PA gene from the pandemic influenza A H1N1/2009 virus (SD07).

**Table 2 viruses-11-00040-t002:** Virus replication ability in the lung tissue of the SPF chickens.

Virus	Dose (EID_50_) ^a^	Lethality	Virus Titers in Lung ^b^ (lgEID_50_/g(SD))	Seroconversion ^d^ (Positive/Total)
SD01	10^6^	None	4.29 ± 0.41(3/3) ^c^	2/2 ^c^
	10^7^	None	5.31 ± 0.15(3/3)	2/2
	10^8^	None	6.14 ± 0.22(3/3)	2/2
rSD01-PA	10^6^	None	4.17 ± 0.38(3/3)	2/2
	10^7^	None	5.06 ± 0.19(3/3)	2/2
	10^8^	None	5.94 ± 0.31(3/3)	2/2

^a^ The parental strain SD01 and the reassortant rSD01-PA were diluted to 10^6^–10^8^ EID_50_. ^b^ SPF chickens (*n* = 5) were infected with different concentrations of virus (10^6^–10^8^ EID_50_). Lung tissues of three SPF chicken were collected on 5 dpi and ground in PBS. The supernatant diluted by 1000 times was used to inoculate the embryos of nine-day-old SPF chickens, which were placed in 37 °C for 72 h. The allantoic fluid of the chicken embryo was collected under a sterile condition, and the viral titer of each tissue was measured. ^c^ The number of positive/total number was tested for each virus. ^d^ Two SPF chickens were observed for two weeks for signs of pathogenicity, and seroconversion was confirmed by a hemagglutination inhibition (HI) assay.

**Table 3 viruses-11-00040-t003:** Number of SPF chickens infected with a virus in the transmission experiments *.

dpi		SD01			rSD01-PA	
	Inoculated	Direct-Contact	Aerosol-Contact	Inoculated	Direct-Contact	Aerosol-Contact
2	6/10	1/10	0/10	2/10	0/10	0/10
4	9/10	3/10	0/10	5/10	0/10	0/10
6	10/10	7/10	3/10	10/10	5/10	0/10
8	10/10	10/10	7/10	9/10	7/10	0/10
10	8/10	7/10	7/10	2/10	4/10	0/10
12	4/10	5/10	5/10	3/10	2/10	0/10
14	0/10	0/10	0/10	0/10	0/10	0/10

* In each independent experiment, 10 SPF chickens (inoculated) were infected with 10^6^ EID_50_ of virus by eye and nasal inoculation, and then another 10 animals (direct contact) were introduced into the same isolator after 24 h. Ten animals (aerosol contact) were placed separately in another isolator B. Oropharyngeal and cloacal cotton swab samples were constantly collected at 2-day intervals and inoculated in SPF embryonated chicken eggs and detected the presence of avian influenza virus in chicken embryos by the HA-test method to determine whether SPF chickens were infected with an avian influenza virus. The result was expressed as the ratio of infected number/total number of chickens. The results indicated that there was no virus shedding of all aerosol contact animals in the experiment of rSD01-PA.

**Table 4 viruses-11-00040-t004:** Results of antibody titers of the serum of SPF chickens in the transmission experiment (log2 ± SD, *n* = 10).

dpi		SD01			rSD01-PA	
	Inoculated ^a^	Direct-Contact	Aerosol-Contact	Inoculated	Direct-Contact	Aerosol-Contact
7	4.66 ± 1.31	2.86 ± 1.52	0	4.73 ± 0.49	0 *	0
14	6.92 ± 1.03	4.59 ± 0.77	4.35 ± 1.27	5.57 ± 1.27	3.24 ± 1.18 *	0 *
21	7.75 ± 0.81	6.32 ± 1.61	5.93 ± 1.30	7.88 ± 1.01	5.33 ± 0.89	0 *

^a^ In each independent experiment, 10 SPF chickens (inoculated) were infected with 10^6^ EID_50_ virus, and then, another 10 animals (direct contact) were introduced into the same isolator after 24 h. Ten animals (aerosol contact) were placed separately in another isolator B. ^b^ Serum samples were constantly collected for 21 days and determined by haemagglutination/ haemagglutination inhibition (HA/HI)-test with reference to the OIE 2008 standard. * *p* < 0.05 (in comparison with SD01 using SPSS 19.0).

**Table 5 viruses-11-00040-t005:** The results of virus titer test in guinea pigs.

Average Virus Titers ^a^ lgTCID_50_/g ± SD						Seroconversion (Positive/Total)
Strain	Brain	Trachea	Nasal Turbinate	Lung	Kidney	
SD01	0 ^b^ (0/3) ^c^	0 (0/3)	0 (0/3)	0 (0/3)	0 (0/3)	0/2
rSD01-PA	0 (0/3)	3.76 ± 0.64 * (2/3)	4.32 ± 0.48 (3/3)	0 (0/3)	0 (0/3)	2/2

^a^ Five guinea pigs were inoculated with 10^6^ EID_50_ virus in a volume of 300 μL. The brains, turbinates, tracheae, lungs, and kidneys tissues of three guinea pigs were collected from each group on 5 dpi. ^b^ Each sample was ground after adding 1 mL sterilized PBS. The supernatant after centrifuge was 10-time diluted and inoculated to the MDCK cells. The viral titer was determined in each tissue by a 50% tissue culture infective dose detection. ^c^ The number of positive/total number tested for each tissue of each virus. * *p* < 0.05 (in comparison with SD01 using SPSS 19.0).

**Table 6 viruses-11-00040-t006:** Number of guinea pigs infected with rSD01-PA in transmission experiments.

dpi	Inoculated ^a^	Direct-Contact	Aerosol-Contact
2	0/5	0/5	0/5
4	3/5	2/5	0/5
6	5/5	5/5	0/5
8	4/5	3/5	0/5
10	4/5	3/5	0/5
12	3/5	3/5	0/5
14	0/5	0/5	0/5

^a^ In each independent experiment, five guinea pigs (inoculated) were inoculated with 10^6^ EID_50_ virus in a volume of 300 μL. Afterward, another five guinea pigs (direct contact) were introduced into the same isolator after 24 h. Five guinea pigs (aerosol contact) were placed separately in another isolator B.

**Table 7 viruses-11-00040-t007:** Results of antibody titers of guinea pigs after inoculating rSD01-PA ^a^ (log2 ± SD, *n* = 5).

dpi	Inoculated	Direct-Contact	Aerosol-Contact
7	5.25 ± 1.7	0	0
14	6.03 ± 0.35	5.32 ± 0.40	0
21	8.35 ± 1.24	7.28 ± 0.73	0

^a^ Five guinea pigs of each group were observed for 21 days, and then the serum was collected for determining antibody titer by the HA/HI method.

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
