# Peer review of "The PA Subunit of the Influenza Virus Polymerase Complex Affects Replication and Airborne Transmission of the H9N2 Subtype Avian Influenza Virus"

_viruses, 2019, doi:10.3390/v11010040_

Reviewer 1 Report

The authors evaluated the role of the PA protein in the replication and airborne transmission of the H9N2 subtype AIV by generating a reassortant H9N2 virus containing the PA protein of a pandemic H1N1 strain. This study showed that the reassortant virus replicated more efficiently in guinea pigs than chickens and gained transmissibility between guinea pigs and guinea pigs by direct-contact. Although the results of this study is interesting, the authors did not include parental H1N1 virus of PA protein for animal study. There is no information available for replication and transmissibility of pandemic H1N1 strain used in this study to elucidate the contribution of the PA protein to viral replication in a mammalian host. It is well-known that influenza replication and pathogenicity are multifactorial. Changing one protein from human influenza virus would not provide sufficient information to elucidate replication and transmissibility of H9N2 virus in a mammalian host. It is not clear whether this specific reassortant event can occur under certain circumstance.

Specific comments

The title is misleading, since this study did not find any aerosol transmission of virus in guinea pigs.

Line 35: include segmented into this sentence.

Lines 256: it is not clear what “specific expression of the receptor” means.

Line 270: change “detoxification began” to “viral replication was detected.”   

Lines 283-298: the authors can combine 3.5.2 and 3.5.3 together into 3.5.2.

Lines 352-368: It has been well-known that H9N2 viruses possessed several genetic markers (i.e., HA H191, HA L234; H9 numbering), which can increase transmission to mammalian hosts. The authors should describe whether the H9N2 strain used in this study possesses any genetic markers.

Author Response

Dear reviewer,

 thank you very much for your professional advice. I have replied to the review comments one by one.

Reviewer 1

I agree with your point of view very much. As you said, influenza replication and pathogenicity are multifactorial. Therefore, we hope to continue to do in-depth research on key sites affecting the pathogenicity and transmissibility of influenza virus after that. Based on your professional advice, I made the following reply

Specific comments

The title is misleading, since this study did not find any aerosol transmission of virus in guinea pigs.

Response: Thank you for your professional suggestion .In this studyThe original virus SD01 was originally airborne among SPF chickens, but the recombinant strainrSD01-PA lost the ability of airborne transmission among SPF chickens. Neither the parental strain nor the recombinant strain has the ability of airborne transmission between guinea pigs, but the recombinant strain (rSD01-PA) has the ability of direct contact for transmission. Therefore, I think the original title is in line with the content of this study.

Line 35: include segmented into this sentence.

Response: Thank you for your professional suggestion and we agreed with your proposal very much. Influenza virus belongs to the orthomyxoviridae RNA virus family, which contains six to eight segmented of linear, negative-sense, single-stranded RNA.

Lines 256: it is not clear what “specific expression of the receptor” means.

Response: Thank you for your professional suggestion. we changed “specific expression of the receptor” to “However, viral replication was not detected in the lungs, which could be related to the distribution of different sialic acid receptors.” What the authors want to express is that no virus has been detected in lung tissues, because the distribution of different sialic acid receptors in different tissues is different.

Line 270: change “detoxification began” to “viral replication was detected.”

Response: Thank you for your professional suggestion and we agreed with your proposal very much. We have changed “detoxification began” to “viral replication was detected.”

Lines 283-298: the authors can combine 3.5.2 and 3.5.3 together into 3.5.2.

Response: Thank you for your professional suggestion and we agreed with your proposal very much. We combined 3.5.2 and 3.5.3 together into 3.5.2.

Lines 352-368: It has been well-known that H9N2 viruses possessed several genetic markers (i.e., HA H191, HA L234; H9 numbering), which can increase transmission to mammalian hosts. The authors should describe whether the H9N2 strain used in this study possesses any genetic markers.

Response: The amino acid sequence of SD01 at HA cleavage site is RSSR↓GLF, and the strain is a low pathogenic avian influenza virus. At HA receptor binding site, the strain has Q-234 (226 H3 number) locus, indicating that it can bind to avian influenza-like sialic acid receptor and infect birds.

Reviewer 2 Report

The authors described that a substitution of PA segment of a chicken H9N2 subtype influenza virus to that of swine H1N1 virus altered the ability of the virus. This substation destroyed the ability of airborne transmission among chickens, but provided infectivity to guinea pigs (mammals). Although the mechanism of this features was not elucidated, this study has shown interesting finding in sight of the host range of avian influenza viruses. To improve the quality of the present study, could you please consider following comments.

Line 208 and 270: The term “detoxification” which the authors likely use for virus shedding is confusable. Please consider to use more direct word.

Line 218: What is “infected number”? The authors should explain how the animals were determined infected or not infected.

Line 253: Please show the unit of viral titers. 

Lines 279-282: Is this footprint of Table 6?

Line 273: What do you mean “virulence continued”? The authors should define “virulence”.

Lines 104, 151, Table 1, Table 2 and Table 5: lgEID50 or lgTCID50 should be revised to logEID50 or logTCID50, respectively.

Line 351: g0.0uinea pigs is typo.

Author Response

Dear reviewer, 

thank you very much for your professional advice. I have replied to the review comments one by one.

Line 208 and 270: The term “detoxification” which the authors likely use for virus shedding is confusable. Please consider to use more direct word.

Response: Thank you for your professional suggestion. Another reviewer raised the same question. We refer to your comments and those of another reviewer. we changed “detoxification” to “viral replication was detected.”

Line 218: What is “infected number”? The authors should explain how the animals were determined infected or not infected.

Response: Thank you for your professional suggestion. “infected number” refers to the number of chickens infected with avian influenza virus. The authors collected cotton swabs from the mouth and cloaca of SPF chickens in isolator, inoculated SPF chicken embryos aseptically, and detected the presence of avian influenza virus in chicken embryos by HA-test method to determine whether SPF chickens were infected with avian influenza virus.

Line 253: Please show the unit of viral titers.

Response: Thank you for your professional suggestion. We have added the unit of viral titer:log10TCID50/gram

Lines 279-282: Is this footprint of Table 6?

Response: Thank you for your professional suggestion. We are very sorry for our incorrect writing. It's a footnote to Table 6. We have modified the footnote format.

Line 273: What do you mean “virulence continued”? The authors should define “virulence”.

Response: Thank you for your professional suggestion. “virulence continued” means “virus shedding”, and changed this part in the manuscript.

Lines 104, 151, Table 1, Table 2 and Table 5: lgEID50 or lgTCID50 should be revised to logEID50 or logTCID50, respectively.

Response: Thank you for your professional suggestion. lgEID50 or lgTCID50 had been revised to logEID50 or logTCID50, respectively.

Line 351: g0.0uinea pigs is typo.

Response: Thank you for your professional suggestion. We are very sorry for our incorrect writing. we changed the” g0.0uinea pigs” to “guinea pigs”

Round  2

Reviewer 1 Report

The authors addressed accordingly.